# Realtime optical flow estimation on vein and artery ultrasound sequences based on knowledge-distillation

Till Nicke[14][], Laura Graf[2][0000−0002−3587−5590], Mikko Lauri[1][0000−0002−2223−9253], Sven Mischkewitz[3][], Simone Frintrop[1][0000−0002−9475−3593], and Mittias P. Heinrich[2][0000−0002−7489−1972]

[1] Department of Informatics, University of Hamburg, Hamburg, Germany
{till.nicke, mikko.lauri, simone.frintrop}@uni-hamburg.de
[2] Institute of Medical Informatics, University of Lübeck, Germany
{graf,heinrich}@imi.uni-luebeck.de
[3] ThinkSono GmbH
sven@thinksono.com
[4] Fraunhofer Institute for Image Computing MEVIS, Lübeck, Germany

**Abstract.** In this paper, we propose an approach for realtime optical flow estimation in ultrasound sequences of vein and arteries based on knowledge distillation. Knowledge distillation is a technique to train a faster, smaller model by learning from cues of larger models. Mobile devices with limited resources could be key in providing effective point-of-care healthcare and motivate the search of more lightweight solutions in the deep learning based image analysis. For ultrasound video analysis motion correspondences of image contents (anatomies) have to be computed for temporal context and for real time application, fast solutions are required. We use a PWC-Net's optical flow estimation output to create soft targets to train a lightweight optical flow estimator. We analyse how well it works on the challenging task of fast segmentation propagation of vein and arteries in ultrasound images. Experiments show that even though we did not fine-tune the teachers on this task, a model trained with soft targets outperformed a model trained directly with labels and without a teacher.

**Keywords:** knowledge distillation · realtime video inference · ultrasound images

## 1 Introduction

The analysis of objects in a sequence of images is a task that plenty of research has been done for, recently mostly in the deep learning field [1]. To achieve a coherent and accurate result over the different time points, it is important that the analysis of the current image considers the past. One form of this temporal context is the estimated optical flow. However, the classical methodology for its' calculation is an iterative approach [2] too slow for realtime inference.

Most recent image registration approaches based on deep learning (e.g. [3]) are computationally too expensive to be executed on mobile devices in the required time. Realtime estimation of optical flow of ultrasound sequences would be advantageous in many practical point-of-care ultrasound (POCUS) applications that are based on intelligent guidance through image analysis. The aim of this work is to train a network, that learns from larger, pre-trained flow estimation networks and is able to accurately propagate relevant information (e.g. segmentations of important anatomies) in ultrasound. Ultrasound images often exhibit ambiguous structure depiction and a network, that employs only 2D convolution without temporal context, is not able to perfectly interpret the image with satisfying accuracy. So instead utilising the motion of the images in a network trained e.g. to propagate the anatomical labels correctly (which is usually coined weakly-supervised registration [4]) can leverage temporal context without requiring access to the whole temporal sequence. Clinically, this is relevant e.g. in the application of labeling vessels for an examination of the leg to diagnose whether a deep vein thrombosis (DVT) is present.

## 2   Related Work

### 2.1   Dynamic ultrasound analysis

The use of automated image analysis for ultrasound is constantly increasing both in research and practical clinical translations [5]. The recent MICCAI challenge CLUST [6] has studied the quality of image registration algorithms for tracking ultrasound but without realtime constraints. A Siamese network for respiratory motion estimation on ultrasound images has been proposed by Liu and colleagues [7], which is capable of tracking landmarks through a video sequence.

A system for compression-based DVT examination in ultrasound (US) images was proposed by Tanno and colleagues [8]. The system, named AutoDVT, uses a dual-task network to help make predictions about the patient's VTE status. One of the tasks consists of classifying the compression status of a registered vein as either closed or open. The network itself uses stacked consecutive frames as input to create temporal consistency. The different task networks share the majority of convolutional layers and only separate the two tasks in the last convolutional layer, thus each task regulates the other during training.

To achieve higher temporal consistency and capture a more holistic view of dynamic sequences, optical flow estimations between frames can be leveraged. To ensure fast inference time, it is of importance that the optical flow prediction takes as little time as possible, while still generating accurate estimations.

### 2.2   Optical flow estimation

In recent research in deep learning and optical flow estimation numerous capable network solutions have been proposed, including Flownet [9], its evolution Flownet2 [10] and PWC-Net [11]. Flownet uses CNN feature extractors on two

images, correlates these features over a discretised displacement search window (originally $21 \times 21$ pixels with a stride of 4), and further processes these correlations to predict a flow field. Flownet2 extends the original Flownet approach by employing multiple different and fine-tuned versions of this architecture.

PWC-Net, which was proposed by Sun et al. [?], on the other hand, uses pyramidal images with a combination of a cost-volume layer and a warping layer to estimate the optical flow of the input images.

In the medical domain LapIRN [12] and PDD-Net [13] are two capable networks for estimating large deformations.

PDD-Net utilizes deformable convolution layers for feature extraction, which are then correlated. The correlation layer is followed by a min convolution and mean-field inference to predict dense displacement probabilities in volumes.

Some of these networks are larger, with up to  162m parameters and up to 0.6 seconds of inference time on an NVIDIA graphics card [9, 10]. However, these models are very accurate, which makes them valuable teachers in a student-teacher setting. Other models, such as the PDD-Net, can be compressed to use little space and computation.

## 2.3   Knowledge Distillation

Student-teacher learning, also known as knowledge distillation, was proposed by Li et al. [14]. The method uses one (or more) large and accurately trained neural network(s), also called teacher, and tries to teach the output distribution to a smaller network, also called student, by minimizing the KL divergence between the teacher's output and the students' prediction.

Yuan et al. proposed that not only accurate teachers can be used in a knowledge distillation setting. In [15] they found that also insufficiently trained teachers can increase the performance of the students, as they provide a representative distribution of the classes in the classification task. Thus, the teachers not only provide accurate information about the output but also provide regularized soft targets.

In [16] Kim et al. compared the KL divergence as a loss function, which is widely used in knowledge distillation, to a mean squared error loss and found, that the mean squared error loss is superior to the KL divergence, especially, when using a small tau, as the label noise is mitigated.

## 2.4   Contributions

We explore one of the aforementioned methods, named knowledge distillation [14] and train a small and lightweight optical flow estimator network for ultrasound motion estimation and vessel segmentation propagation in ultrasound images. We also compare this method to a different training setup to evaluate the usage of the distilled knowledge and find an increase in Dice score, as well as a decrease in Hausdorff distance (HD). As segmented medical reference data

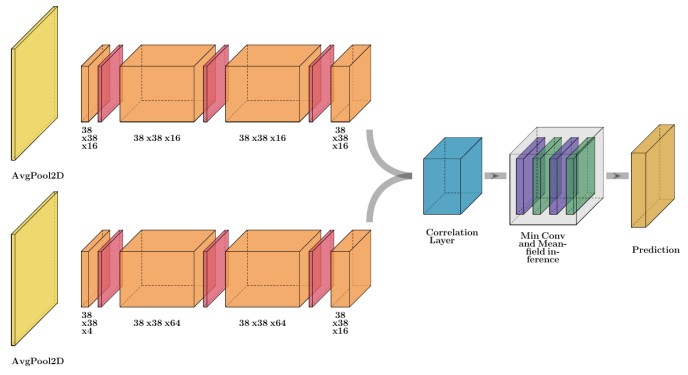

**Fig. 1.** Overview of the PDD-Network architecture for image registration, which comprises deformable convolutions with batch normalisation and ReLU (red), a correlation layer (blue) and differentiable mean-field inference as regularisation (purple and green).

is scarce, this approach could potentially help increase performances for ultrasound image processing.

We aim at a short inference time of the optical flow to either create an additional input for further image analysis networks or to use the optical flow itself for segmentation propagation on mobile devices, such as tablets or phones. This constrains size and throughput of the network, as computational power on mobile devices differs greatly from stationary setups. Therefore, we use a lightweight version of the aforementioned PDD-Net as student.

## 3    Method

We used the PDD-Net [13], which achieved competitive results in the Learn2Reg challenge [17]. A 2D implementation was made available [5] (Figure 1). In this version of the Model, an average pooled (yellow) version of the input image is processed by three convolutional layers each followed by batch normalisation and ReLU (red). After the first convolution, we adapt the 2D implementation by applying an Obelisk layer [18], which is then followed by two more convolutional layers. The Obelisk layer is a form of deformable convolution, which uses learnable weights and a gridsampling operator, to increase the receptive field of the next convolutional layer [18].

For a fixed and a moving image, the extracted features are correlated, akin to the correlation used in Flownet-C [9] and then further processed with min convolutions and mean-field inference [19].

The whole model has an inference time of around 2.7 ms on an Nvidia RTX 2060 Ti GPU. When looking at the model (Figure 1), we can see two feature

---

[5] https://www.kaggle.com/mattiaspaul/learn2reg-tutorial

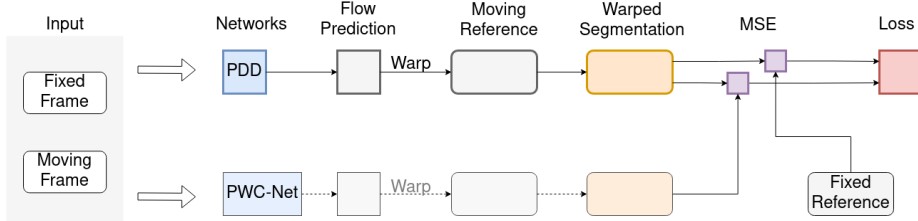

**Fig. 2.** Illustration of our concept for knowledge distillation for DL-based optical flow estimation. The teacher (PWC-Net) was not trained on ultrasound sequences but can provide a soft target for our student (PDD-Net) based on only a single reference frame segmentation.

extractors, which share weights. By processing one fixed frame at time $t$ and keeping this frame as a fixed frame, we only need to process the moving frame at point $t + x$ of the video through the CNN. By reducing the convolutional operations needed during video processing, the network's inference time could be reduced to 1.7 ms.

The same optimization can be applied when using different fixed images. In that case the extracted feature map of the moving frame can be re-purposed as feature map of fixed frame, when a new moving frame is presented.

 The PDD-Net is trained on a combination of soft and hard targets. The hard target loss is calculated as the MSE between the one-hot encoded reference segmentation ("fixed reference" in Figure 2), and the networks' prediction. The prediction is generated by using the predicted flow field to warp the reference segmentation from the moving frame towards the fixed frame. This warped segmentation is then compared to the reference segmentation of the fixed frame.

We use the established optical flow estimator PWC-Net [11] as a teacher to provide soft targets during training. This is done as shown in Figure 2. To generate the soft targets, the PWC-Net's optical flow prediction is used to warp the reference segmentation of the moving frame towards the fixed frame. We calculate the MSE loss between the one-hot encoded warped moving reference segmentations of teacher and student networks. The soft and hard target loss and then summed up, where the soft target loss is scaled by 0.5.

**Experimental setup:**  The task is to propagate a single reference segmentation if veins and arteries through a video of about 10 seconds of a DVT examination. The dataset was provided by ThinkSono GmbH [6]. The dataset contains video sequences of DVT examinations that were annotated by experts. An overlay of these reference segmentations can be seen in Figure 3. We use 250 video IDs to create two datasets with which we capture two distinct properties. The first datset is created as a training dateset an contains 1743 image pairs with a fixed frame distance of 6 frames that were randomly sampled. Thus, cap-

---

[6] https://thinksono.com/

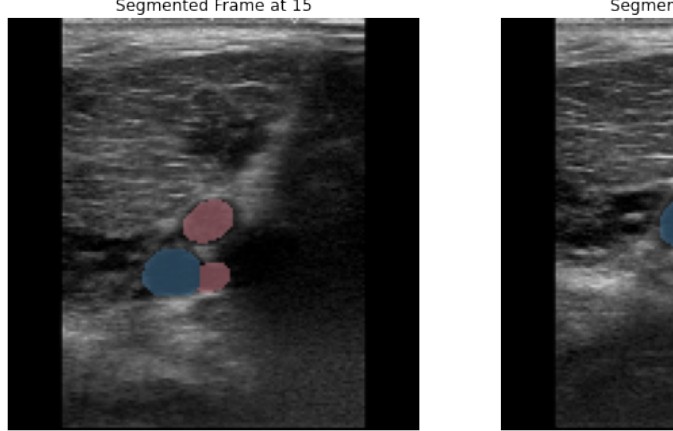

**Fig. 3.** Exemplary image pair used in the fine tuning data set. Reference segmentations are added for better visualization.

turing smaller and larger displacements while also providing heterogenous image quality. The second dataset is created as a fine-tuning dataset. This dataset is created to provide task specific data. We select one random frame in the first fifth of the video, or before the onset of the vein compression (whatever came first), for every ID. We then sample the coming frames with a frame distance of 4 and create image pairs with one fixed frame and various moving frames, resulting in 3285 image pairs. In this dataset larger displacements and compressions of the vein are captured.

We proceed to train the PDD-Net adaptation on the training data set with additional soft targets from the PWC-Net (2) over 100 epochs with a learning rate of 0.002 and an Adam optimizer. We then trained the distilled network on the fine-tuning data set for 200 epochs with a learning rate of 0.00025. For comparison, we also train one version of the PDD-Net adaptation without additional soft targets.

## 4    Results and discussion

We evaluated the networks on 23 unseen videos containing approximately 1600 Frames overall. For each video, we selected one random frame in the first fifth of the video, which we refer to as $f_t$, for frame at time point $t$. Each following frame $f_{t+x}$ is used as moving frame input. The estimated optic flow between $f_t$ and $f_{t+x}$ is used to warp the reference segmentation from $t$ to $t + x$, where it is compared to reference segmentation at time pint $t + x$.
This procedure allows us to apply the mentioned runtime optimization towards video processing. By passing the fixed frame once, keeping it in memory for

**Table 1.** Mean Dice ↑ over the test IDs and Mean HD ↓ over the IDs. Comparison between label loss and KD trained PDD-Nets

| Score | registration | | segmentation |
|---|---|---|---|
| | $PDD$ | $PDD_{KD}$ | nnU-Net |
| vein Dice % | $46.9 \pm 4.13$ | $\mathbf{47.92} \pm 4.15$ | $45.93 \pm 6.47$ |
| artery Dice % | $44.48 \pm 6.08$ | $46.67 \pm 6.28$ | $\mathbf{66.80} \pm 6.91$ |
| overall Dice % | $45.69 \pm 5.0$ | $47.3 \pm 5.09$ | $\mathbf{56.36} \pm 7.77$ |
| vein HD | $25.28 \pm 166.82$ | $24.16 \pm 159.5$ | $\mathbf{23.71} \pm 366.06$ |
| artery HD | $28.3 \pm 205.19$ | $27.7 \pm 205.54$ | $\mathbf{26.88} \pm 640.51$ |
| overall HD | $26.79 \pm 183.79$ | $25.93 \pm 181.26$ | $\mathbf{25.33} \pm 508.84$ |

correlation, solely the moving frames need to be passed through the CNN for feature extraction. The reduced inference time per image is about as fast as a reference segmentation network, nnU-Net, which takes 1.6 ms on the same GPU (Nvidia RTX 280Ti).

As mentioned by Reinke [20] there are common limitations when applying only one metric to measure the performance of segmentation masks. Therefore, we evaluated the two networks with the Dice score and Hausdorff distance. The dice score is used as a measurement of overlap between the reference and predicted segmentation. It ranges from 0 to 1, where 1 is the best score, which we have denoted by ↑. The HD is used as a measurement of furthest distance between reference and predicted segmentation. We show the absolute values, where lower is better, as denoted by ↓. The mean results over all IDs can be seen in Table 1.

We found the distilled network to perform slightly better compared to the label loss trained network over both metrics. When looking at the dice score between the two networks, we found a 2% increase in accuracy over artery segmentation and a 1% increase in vein segmentation. When looking at the HD, we found a similar pattern. The KD trained network outperforms the label loss trained network slightly. We argue that this slight increase is due to the different conceptual representation learned by the distilled network, which would be in line with current research [14, 16, 21]. The PWC-Net scored at $40.56 \pm 3.74$ in dice and $26.51 \pm 160.42$ in HD on the evaluation videos.

When compared to a 2D segmentation network (nnU-Net [22] Table 1), which was trained on the same image IDs, as the optical flow estimator, we find that the distilled network is performing slightly worse in HD, and worse in Dice score. This result is somewhat expected, since the motion during longer sequences can have significant deformations (compression ultrasound of veins) and substantial drift. The frame-by-frame segmentation is in principle translation invariant and was trained with a large number of ground truth segmentation annotations. However, when visually looking at estimated segmentations (and quantitatively the variance in HD between the optical flow method and the nnU-Net), we can see that the segmentation network has limited temporal consistency. This suggests that the 2D nnU-Net creates less smooth segmentations over a video, compared to the optical flow method. In the future, we therefore plan to experiment with

the optical flow as additional input for a segmentation network. Using a deformation field between two frames, instead of a stacked tensor of all frames, can reduce the computational effort needed for processing, while at the same time containing almost as much information as stacked consecutive frames.

Especially during compression of the vein, this additional information can be leveraged. Figure 4 shows the estimated deformation field between the fixed and the moving frame. The compression is clearly visible as and marked with a black bounding box.

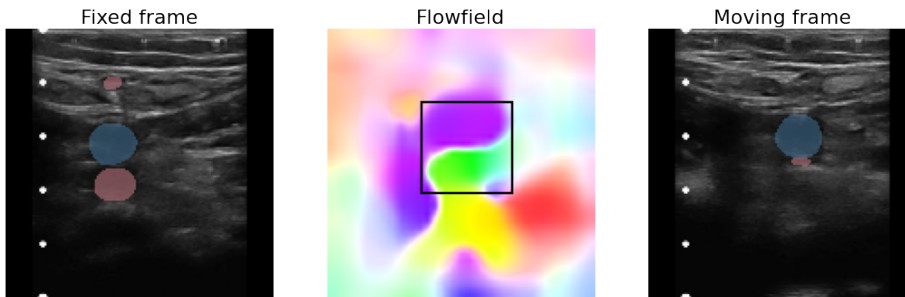

**Fig. 4.** Visualised deformation field between fixed and moving frame. Segmentation was overlayed for better visibility. The bounding box shows where the compression of the vein (pink) is located and in which direction the vein is compressed.

## 5   Conclusion

As part of a Masters thesis, we presented experiments on possible benefits of cross-domain knowledge distillation (from computer vision to medical imaging) for training an optical flow estimator, in this paper. By using additional teacher-generated soft targets during training, we were able to achieve a small increase in Dice score and a small decrease in Hausdorff distance. This shows that cross-domain KD can have a beneficial effect applied in the training of an image registration network.

We were able to adjust our approach to video inference, such that it is capable of running in realtime, with 1.7 ms per frame pair or more than 500 frames per second. Estimating our approach to use approximately 0.14 GFlops per image, we can calculate an upper limit of roughly 230 frames per second on modern mobile GPUs (Qualcomm Adreno 660).

Performance of segmentation networks still exceeded segmentation via this optical flow based registration of the labels. But we suggest an increase in the segmentation networks' accuracy is possible by combining optical flow information with image features, to add temporal context to the segmentation formation. This was already suggested in previous research in medical video segmentation

[23], where improved temporal coherence is reported when optical flow is incorporated. Therefore, we will further investigate the influence of optical flow on vessel segmentation in ultrasound videos.

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
