# OpenReview forum: "Real-time optical flow estimation on vein and artery ultrasound sequences based on knowledge-distillation"
_WBIR.info/2022/Workshop/Biomedical_Imaging_Registration — WBIR 2022_

### Official Review · Reviewer_9TKz · 2022-02-11

**Rating:** 5
**Confidence:** 4
**Recommendation:** Long Oral

**Deanonymize Review:**

no

**Detailed Comments:**

-	references PWC-Net are not properly processed
-	p6 keeping in in memory -> keeping it in memory
-	p6 can be seen in 1 -> can be seen in Table 1
-	ref 5 is incomplete.


**Paper Type:**

methodological development

**Strengths Weaknesses:**

The authors propose a fast approach for motion correction and segmentation propagation in ultrasound sequences. The registration is based on optic flow and uses a lightweight network, trained in a teacher-student setup. The method is evaluated on a substantial set of clinical images. A comparison to a direct segmentation network is included.

Strengths:
-	interesting approach to achieve a solution for point-of-care image analysis
-	comparison to state-of-the-art technique
-	good inference speed

Weaknesses:
-	results promising, but not yet competitive

The approach suggested is a sensible one. Performance is improved slightly, but I assume these are first results and further advances can be found in future. I applaud the authors for including the comparison with the nnU-Net and openly concluding that the performance of the proposed method is not quite at the same level (yet).
I understand the motivation to select a single frame as the fixed frame to register all subsequent frames to. However, a temporally smoother result is expected when consecutive frames would be registered. Moreover, optic flow is an approach more suited for temporal sequences with small transformations between consecutive frames. By changing the problem to registration of frames that are distant (in a temporal sense), optic flow may not be the optimal choice.
Some information on the data used is missing. What were the acquisition parameters and how were the annotations created? Were the data used for testing from the same set or did the acquisition differ?
Were both data sets (those with a fixed frame distance and those with varying distances) used for training? What is the motivation for including the former?

---

### Official Review · Reviewer_BkuS · 2022-02-17

**Rating:** 4
**Confidence:** 3
**Recommendation:** Short Oral

**Deanonymize Review:**

no

**Detailed Comments:**

I find the paper well-written, the application clear, and it was generally an interesting read. The paper does not present major breakthroughs, but I think it can be of interest to the WBIR audience and I therefore recommend acceptance.

**Paper Type:**

validation / application paper

**Strengths Weaknesses:**

The paper concerns optical flow prediction on ultrasound sequences, specifically knowledge distillation where a student network is trained to mimic a teacher network. The aim is to reduce computational complexity to allow the flow estimation to run in real-time on mobile devices.

Strengths:
* the goal of allowing real-time usage on mobile devices can have impact for applications in the clinic
* the paper is well-written and the presentation clear
* the authors reach their aim of reducing computational complexity while achieving comparable scores to the larger teacher networks

Weaknesses:
* there is no major methodological contribution in the paper

---

### Official Review · Reviewer_Zti1 · 2022-02-21

**Rating:** 3
**Confidence:** 4
**Recommendation:** Short Oral

**Deanonymize Review:**

no

**Detailed Comments:**

The idea to use knowledge distillation for optical flow estimation is interesting. However, I have several concerns, mainly about the application for this task for ultrasound segmentation, the model description and the rather limited evaluation, which prevents me from recommending to accept the paper.

Here some additional/minor comments:
- Sec 3: At first it remains unclear what the authors mean by “By applying some optimization for video processing, we can reduce this time to 1.7 ms.” Only later it is described that the same fixed image is used for all estimations of one sequence. Like this they avoid the feature extraction of the fixed image for each image pair.
Even if the estimation of the optical flow for a sequence is faster like this, I doubt that this is a good strategy. The anatomical structures can change dramatically between several frames and I would assume that the estimation is more accurate between adjacent frames.
- When referencing the figures/tables, the word ‘Figure’/’Table’ is missing, which is confusing.
- Fig. 1: A more detailed caption is needed, including labeling of all components.
- Fig. 2: the caption and description in the text is not sufficient. Why are there two fixed images (fixed frame and fixed reference)? I don’t understand the difference between them.
- PWC-Net reference is incorrect ([?])
- Please remove the dot after ‘et’ in ‘et al.’
- Table 1: the caption should be above the table; Second-last row: extra 2 in PDD_KD column

**Paper Type:**

both

**Strengths Weaknesses:**

Strengths
- The paper explores knowledge distillation for optical flow estimation in ultrasound sequences. As a teacher they use PWC-Net, a state-of-the-art model for optical flow estimation in computer vision. It is relatively heavy and the idea to distillate its ‘knowledge’ in a light weight student model, which could potentially operate on a mobile device, is valid.
- Overall, the paper is well-structured, and the related work is described sufficiently.

Weaknesses
Unfortunately, the paper has several weaknesses.
- If I understood it correctly, the main task of the pipeline is to perform segmentation in a temporal sequence of US images. How meaningful are the registrations when aligning such images? The structure is changing over time when sweeping through, so the registered structures may not correspond anatomically in an image pair. It seems to me that for this problem a segmentation model taking into account the temporal dimension might be a better approach than registration. A discussion on this is missing.
- Some parts of the method are not clear. Fig. 2 suggests that the knowledge distillation is obtained using the MSE loss of segmentation labels only. What is the regular MSE loss of the student?
What is the difference between the fixed frame and the fixed reference in Fig. 2?
The authors describe some aspects (choice of segmentation measures, Obelisk layers, long section on related work) in detail, but neglect the essential parts of the paper.
- The data needs a more detailed description. How is the US sequence acquired? Who performed the reference annotations? How many videos/images were used for training?
It is not clear, why two datasets are constructed and how many images/image pairs are in each dataset.
- The authors only evaluated the segmentation performance. But how good is actually the optical flow measurement? How effective was the knowledge distillation? How plausible are the estimated flow fields? The authors do not show any visual results, also not for the segmentations. A Dice of 50% and HD of 25 mm doesn’t seem to be very good. How successful was the label propagation actually?
- PWC-Net is used as a teacher model. How good is PWC for the task at hand? It would be important to compare those results. Otherwise it is unclear why PWC should be used for knowledge distillation.

---

### Decision · Program_Chairs · 2022-02-22

Accept